# Infrared Spectroscopy of Synovial Fluid Shows Accuracy as an Early Biomarker in an Equine Model of Traumatic Osteoarthritis

**DOI:** 10.3390/ani14070986

**Published:** 2024-03-22

**Authors:** Luca Panizzi, Matthieu Vignes, Keren E. Dittmer, Mark R. Waterland, Chris W. Rogers, Hiroki Sano, C. Wayne McIlwraith, Christopher B. Riley

**Affiliations:** 1School of Veterinary Science, Massey University, Palmerston North 4442, New Zealand; l.panizzi@massey.ac.nz (L.P.); c.w.rogers@massey.ac.nz (C.W.R.); 2School of Mathematical and Computational Sciences, Massey University, Palmerston North 4442, New Zealand; m.vignes@massey.ac.nz; 3School of Natural Sciences, Massey University, Palmerston North 4442, New Zealand; m.waterland@massey.ac.nz; 4School of Agriculture and Environment, Massey University, Palmerston North 4442, New Zealand; 5Veterinary Specialty Hospital Hong Kong, G/F—2/F 165-171 Wan Chai Road, Wan Chai, Hong Kong, China; hiloki50@hotmail.com; 6Orthopaedic Research Center, C. Wayne McIlwraith Translational Medicine Institute, Colorado State University, Fort Collins, CO 80523, USA; wayne.mcilwraith@colostate.edu; 7Department of Clinical Studies, Ontario Veterinary College, Guelph, ON N1G 2W1, Canada

**Keywords:** osteoarthritis, biomarker, horse, equine, infrared, spectroscopy, synovial fluid, carpal, traumatic, model

## Abstract

**Simple Summary:**

Osteoarthritis is a leading cause of lameness and joint disease in horses. A simple, economical, and accurate diagnostic test is required to routinely screen horses for OA. The study assessed the accuracy of infrared (IR) spectroscopy in analyzing synovial fluid (SF) to identify horses with early inflammatory changes related to equine carpal osteoarthritis (OA). OA was surgically induced in one group of horses while the others were allocated as controls. SF samples were collected before OA induction and weekly until 63 days. IR spectroscopy was used to analyze the SF samples, and predictive models were created to classify the samples. Overall, the accuracy for distinguishing between joints with OA and any other joint was 80%. Results show that IR spectroscopy could classify samples based on the day they were collected with 87% accuracy. Distinguishing between OA vs. OA Control and OA vs. Sham joints had lower accuracies of 75% and 70%, respectively. The authors conclude that IR spectroscopy accurately discriminates between SF in joints with induced OA and controls.

**Abstract:**

Osteoarthritis is a leading cause of lameness and joint disease in horses. A simple, economical, and accurate diagnostic test is required for routine screening for OA. This study aimed to evaluate infrared (IR)-based synovial fluid biomarker profiling to detect early changes associated with a traumatically induced model of equine carpal osteoarthritis (OA). Unilateral carpal OA was induced arthroscopically in 9 of 17 healthy thoroughbred fillies; the remainder served as Sham-operated controls. The median age of both groups was 2 years. Synovial fluid (SF) was obtained before surgical induction of OA (Day 0) and weekly until Day 63. IR absorbance spectra were acquired from dried SF films. Following spectral pre-processing, predictive models using random forests were used to differentiate OA, Sham, and Control samples. The accuracy for distinguishing between OA and any other joint group was 80%. The classification accuracy by sampling day was 87%. For paired classification tasks, the accuracies by joint were 75% for OA vs. OA Control and 70% for OA vs. Sham. The accuracy for separating horses by group (OA vs. Sham) was 68%. In conclusion, SF IR spectroscopy accurately discriminates traumatically induced OA joints from controls.

## 1. Introduction

Osteoarthritis (OA) is a significant cause of lameness and joint disease in horses [1,2], with approximately 50% of United States multi-horse operations experiencing one or more lameness cases yearly, of which half are limb- or joint-related [3].

There is a need to identify biomarkers of OA to identify early disease onset, stratify the likelihood of progression, and explore the response to new interventions [4,5,6]. The term biomarker has been defined as an objectively measured indicator of homeostatic biological processes, pathologic changes, or responses to treatment [6]. The indicators include “any substance, structure, or process that can be measured in the body or its products and influences or predicts the incidence or outcome of disease” [7]. Several soluble biomarkers of joint metabolism and disease have been explored for their use as potential human and animal OA markers in synovial fluid (SF), blood, and urine [5,6,8,9,10]. Although significant progress has been made in equine biomarker research [11,12,13,14], and there is evidence of potential clinical application for those based on serum [15] and synovial fluid [15,16,17], validation and widespread clinical use of biomarkers has not become common practice [9]. Multiple factors contribute to this lack of uptake, including high costs, low practicality, and inconsistency of results for disease surveillance [18]. 

Obtaining an early diagnosis of OA remains a challenge, and it is difficult to clinically assess disease progression and the response to therapeutic interventions [9]. Hence, cost-effective and reliable testing methodologies are needed, especially in the early stages of OA when clinical signs are absent in at-risk populations [19,20]. Unlike in humans [21], clinical staging of disease and case definitions of OA for specific joints have not been established in animals. This is viewed as essential in the monitoring of OA and the validation of treatments [6]. However, a significant body of knowledge has been gained from animal models, including dogs [19] and horses, especially for the early stages of the disease [14,22,23,24,25,26]. Proteomic approaches have been developed recently using blood, synovial fluid, and other body fluids to investigate their use in OA for diagnostic, prognostic, and therapeutic purposes [27]. These more costly techniques rely on protein separation and characterization by mass spectrometry. 

An alternative approach to screening for equine OA is based on identifying a biochemical profile of multiple molecules in films of microliter-sized samples of biological fluids using Fourier transform infrared (IR) spectroscopy [28]. This technique does not require the separation of single molecular species associated with disease but instead provides a complex IR signal produced by an array of molecules [29]. Since OA is a complex disease triggered by trauma or other causes of inflammation affecting multiple joint tissues, IR spectroscopy has been investigated to evaluate a range of known and unknown biomarkers simultaneously [30,31]. Moreover, as biomarker expression changes in character, or waxes and wanes with natural disease [6], a consistent diagnostic platform can be used [28]. In human joints affected by rheumatoid arthritis, IR spectroscopy of patient SF has shown potential as a diagnostic and screening tool [30,31,32]. Similarly, IR spectroscopy of SF from equine clinical cases has been demonstrated to capably identify osteochondrosis and naturally occurring traumatic arthritis [16,17]. In the description of equine traumatic arthritis in a mixed population of racehorses with clinical signs presented for surgical treatment, accuracy (97%), sensitivity (93%), and specificity (100%) were high [16]. In canine OA associated with cranial cruciate ligament rupture, IR spectroscopy of synovial fluid allowed differentiation between affected and control joints with high sensitivity (97.6%), specificity (99.7%), and overall accuracy (98.6%) [33]. However, in both studies, the timing of the onset of early disease and clinical signs and presentations was unknown. The advantages of IR spectroscopy are its accuracy, low cost, and low invasiveness [28]. Although this technique has performed well in SF from horses with naturally occurring diseases, there are no studies assessing its use for the early detection of OA in a controlled research setting using a homogenous cohort of horses. This is a necessary validation step before considering this approach as a tool for preclinical screening of at-risk horses or assessing joint disease responses to therapeutic interventions that might prevent the need for surgical intervention [6,28,34].

The objective of this study was to evaluate IR-based SF biomarker profiling to differentiate joints with early inflammatory changes associated with a well-established traumatically induced model of equine carpal osteoarthritis (OA) from controls. The authors hypothesized that biomarker profiling via analyses of IR spectra of dried films of SF could differentiate joints with induced carpal OA from controls.

## 2. Materials and Methods

This study was approved by the Massey University Animal Ethics Committee (MUAEC 14/18). The sample size for treatment and Sham Control groups was estimated based on previous work using the carpal chip model for other biomarker studies and treatment trials [22,25,26].

### 2.1. Animals and Surgical Protocol

Seventeen female New Zealand-bred thoroughbred horses including fifteen 2-year-olds and two 3-year-olds were recruited for the IR-based biomarker study. These horses had not previously been trained or used for any athletic activity. To determine eligibility for enrolment in the study, the animals were checked by veterinary examination for physical abnormalities or illness, lameness at walk and trot as assessed by two specialist equine surgeons, carpal flexion tests, and radiographic examination of the carpi; all findings were negative. Procaine penicillin at 22 mg/kg IM (Phoenix Pharmacillin 300, 300 mg/mL, Phoenix Pharm, Auckland, New Zealand) was administered once before arthroscopy. For the assignment into the Sham operation (control horse) or the surgical induction of traumatic OA groups, horses were blocked for sire to minimize genetic relatedness within groups and randomized using the RAND function in Excel (Microsoft Excel 2013, Version 15.0.4727.1003, Microsoft Corporation, Auckland, New Zealand).

Nine horses (median age 2 years; IQR = 0) were randomly assigned to the OA group (i.e., the treatment group). Horses were sedated with 1 mg/kg IV of xylazine (Ceva Thiazine 100 Injection; Ceva Animal Health Pty Ltd., Glenorie, Australia). General anesthesia was induced with 2.5 mg/kg of ketamine HCl (Ceva Ketamine injection; Ceva Animal Health Pty Ltd., Glenorie, Australia) and 0.01 mg/kg of diazepam (ilium Diazepam Injection USP; Troy Laboratories, Glendenning, Australia) intravenously. General anesthesia was maintained with isoflurane (Isoflurane; Bayer New Zealand Ltd., Auckland, New Zealand) in 5 L/min of 100% oxygen. Aseptic preparation of a randomly chosen carpus was followed by middle carpal arthroscopy, and an 8 mm osteochondral fragment was created using a bone gouge in the distal dorsal radial carpal bone [24,25]. The osteochondral fragment was left attached at the reflection of the dorsal joint capsule. The bone of the fracture bed was debrided with a motorized bone burr, creating a ~15 mm-wide defect (inclusive of the fragment’s width). Tissue debris was left in the joint. The skin was closed with simple interrupted 2-0 nylon sutures (Ethilon, Ethicon, Johnson and Johnson New Zealand Limited, Auckland, New Zealand). The incisions were dressed, and the carpal bandaged. For horses identified as members of the OA horse group, their operated middle carpal joints were designated OA joints, and the unoperated contralateral middle carpal joint was designated OA Control joints.

The remaining eight horses (Sham horse group; median age 2 years; IQR = 0) underwent arthroscopic exploration of one randomly selected middle carpal joint using the same general anesthetic protocol without creating an osteochondral defect (Sham joint). The unoperated contralateral middle carpal joint in these horses served as a Sham Control joint. All horses were administered phenylbutazone immediately after completion of the procedure at 4.4 mg/kg IV (Nabudone P, 200 mg/mL, Troy Laboratories, Glendenning, Australia) and for the following four days at 4.4 mg/kg PO every 24 h (Equine Bute Paste, 200 mg/mL, Randlab, Revesby, Australia). Postoperatively, subjects underwent clinical examination twice daily to evaluate their comfort and well-being.

### 2.2. Postoperative Exercise and Clinical Assessment

After a 14-day recovery period in box stalls with 30 min of daily turnout, horses underwent a 7-week-long treadmill exercise protocol (5 days/week). For this, exercise was provided once daily for two minutes at a trot (4–5 m/s), then two minutes at a gallop (8–9 m/s), and finally two minutes at a trot (4–5 m/s). The model has been used previously by other researchers to mimic naturally occurring equine traumatic OA [25,26]. Each horse was assessed pre-intervention and once weekly thereafter to grade lameness [35], joint effusion, and response to carpal flexion. Radiographs were taken on termination of the protocol, and scores were given by veterinary radiologists blinded to treatment groups for radiographic changes to confirm the establishment of OA. Results of the lameness, flexion tests, effusion, and radiographic scores for this study have been previously published [36].

### 2.3. Synovial Fluid Sample Collection

Three mL of synovial fluid (SF) was collected aseptically from both carpi before surgical induction of OA (or Sham surgery) on Day 0 and then weekly from all horses until Day 63. SF aliquots of ~1 mL were stored at −80 °C for later batch analysis. 

### 2.4. Infrared Spectroscopy of Synovial Fluid

Synovial fluid samples were thawed at 20 °C, and replicate (×6) dry films were made for each aliquot on a silicon 96-well microplate [16,17,33,36]. The microplate was mounted on a multi-sampler accessory (XY Microtiter Plate Accessory, PIKE Technologies, Madison, WI, USA) interfaced with an IR spectrometer (Tensor 27, Bruker Optics, Preston, Victoria, Australia). Infrared absorbance spectra of were generated and recorded with proprietary software (OPUS software, version 6.5, Bruker Optics, Ettlingen, Germany). For each sample, 512 IR interferograms were averaged, and Fourier transformed to obtain a spectrum with a resolution of 4 cm^–1^ over the 400 to 4000 cm^–1^ wave number (WN) range.

### 2.5. Analyses of Synovial Fluid Spectral Data 

#### 2.5.1. Spectral Pre-Processing

Spectral files were converted into delimited data CSV files for further analyses using proprietary software (The Unscrambler Xv10.5.1, Camo Software, Oslo, Norway). Subsequent analyses were performed in R (V4.2.2, R Core Team, Auckland, New Zealand) using the prospectr R package V 0.2.4. Savitzky–Golay filtering was applied to all spectra with a 2nd-order derivative of the signal, a 2nd-order polynomial function, and a smoothing window of width of 15; parameters were tuned to maximize spectral separation by day. An analysis of sensitivity to other filtering parameter combinations was performed before selecting this combination of parameters. Spectra were normalized and baseline effects removed by standard normal variate (SNV) transformation, reducing within-class variance [37]. The ”fingerprint” regions between wavenumber ranges 1300–1800 cm^–1^ and 2600–3700 cm^–1^ were selected for further analyses [29]. The number of outliers did not exceed the threshold of extreme PCA scores expected by chance (5%), and so no spectra were excluded.

#### 2.5.2. Classification Model Development

Predictive models were built to predict: (1)The sampling day (task 1; Days 0, 7, 14, 21, 28, 35, 42, 49, 56, and 63; 10 classes);(2)The joint sampled in OA horses (task 2; OA joint vs. OA Control; 2 classes);(3)The joint sampled in Sham horses (task 3; Sham joint vs. Sham Control, 2 classes);(4)The intervention joint sampled between horse groups (task 4: OA joint vs. Sham joint, 2 classes);(5)All joints sampled in both horse groups (task 5a: OA joint, OA Control, Sham joint vs. Sham Control; 4 classes);(6)The OA joint sample vs. any other (task 5b; 2 classes);(7)The horse group (task 6; OA versus Sham; 2 classes);(8)The samples classified day × joint group except for Day 0 (i.e., before interventions) for which OA and Sham groups were pooled (task 7a; Day 0, Day 7 × OA joint, Day 7 × OA Control, Day 7 × Sham joint, Day 7 × Sham Control joint, Day 14 × OA joint, Day 14 × OA Control, Day 14 × Sham joint, Day 14 × Sham Control joint, Day 21 × OA joint, Day 21 × OA Control, Day 21 × Sham joint, Day 21 × Sham Control joint, Day 28 × OA joint, Day 28 × OA Control, Day 28 × Sham joint, Day 28 × Sham Control joint, Day 35 × OA joint, Day 35 × OA Control, Day 35 × Sham joint, Day 35 × Sham Control joint, Day 42 × OA joint, Day 42 × OA Control, Day 42 × Sham joint, Day 42 × Sham Control joint, Day 49 × OA joint, Day 49 × OA Control, Day 49 × Sham joint, Day 49 × Sham Control joint, Day 56 × OA joint, Day 56 × OA Control, Day 56 × Sham joint, Day 56 × Sham Control joint, Day 63 × OA joint, Day 63 × OA Control, Day 63 × Sham joint, Day 63 × Sham Control joint; 37 classes);(9)Similarly comparing the day × OA joint sampled vs. any other (task 7b; 19 classes), and the variation among horses (task 8, horse labels 1 to 17; 17 classes).

Sparse partial least squares discriminant analysis [38], logistic and multinomial regression with L1-regularization [39], random forests [40], Support Vector Machines [41], and convolutional neural networks [42] were explored as classification methods to ensure that model/algorithm assumptions were not performance-limiting. However, overall, random forests allowed for the most efficient detection of the best classification performance and were easy to implement without requiring complex tuning of the method. The random forest v. 4.7-157 and ranger v. 0.15.158 R packages with default parameter settings were used. The number of trees to ensure convergence was explored by monitoring the global accuracy, and it was found that 1000 trees were sufficient for all considered tasks. Doubling and dividing the default number of splits (√(number of WN)) by a factor of two did not change the performance. For unbalanced classification tasks, samples were re-weighted. For example, task 5a included 25% of joint spectra classified as OA versus spectra of the OA Control, Sham, and Sham Control joints that compromised the remaining 75% of IR-based measurements.

Predictions were made for each tree in the forest using out-of-bag (OOB) samples [43]. A confusion matrix was then obtained by comparing the predicted class to the actual class of the OOB samples for all the trees in the forest. The authors report the overall classification rate (computed as the ratio of the sum of the diagonal elements of the confusion matrix to that of the sum of all its elements) for each task and, when relevant (i.e., when imbalanced performance was observed), the classification rate per-class (proportion of sample of one class correctly identified as such). The overall classification rate is commonly referred to as accuracy. A random classification for a balanced problem would lead to a performance rate of 1/(number of classes), so 50% for a 2-class problem, 25% for a 4-class problem, etc. The accuracy of a predictive model is defined as very good (>90%), good (70–90%), acceptable (60–69%), or poor (<60%).

## 3. Results

### 3.1. Spectral Pre-Processing

The raw spectra of SF from all joints by class are shown in Figure 1a (top). No obvious visual distinction between joints was evident in the spectral patterns among groups. The higher peaks of the spectra are associated with intermolecular bond vibrations with proteins; the absorption bands centered at 1650 cm^−1^ (amide I) and 1545 cm^−1^ (amide II) are associated with stretching and bending vibrations of the amide C=O and N–H groups, respectively. Absorption at 3300 cm^−1^ is also associated with the N–H group but is a stretching vibration (amide A mode) [44]. Figure 1b (bottom) shows the image of the pre-processed spectra.

### 3.2. Classification of Synovial Fluid IR Spectra

For classification task 1, separation of spectra based on the day of sampling (Days 10 to 63), the accuracy of classification among all horses was 87.0%, with a very good per-class correct classification rate (over 88.0%), except for Day 56 (51.5%) and Day 63 (52.9%), which the model could not distinguish. The accuracy within the OA horse group (task 2; OA vs. OA Control) was 75.0%, which contrasts with a lower accuracy (61.0%) within the Sham horse group (task 3; Sham vs. Sham Control). For prediction in task 4 (OA joints vs. Sham joints), the accuracy was good (70.0%). The accuracy by all joint groups in all horses (task 5) was 53.0% with OA joints, which were more precisely classified than Sham joints (66.4% and 64.2% for OA Control joints and OA joints, respectively, vs. 37.2% and 40.3% for Sham Control joints and Sham joints, respectively). For task 5b, the accuracy in identifying OA joints vs. any other joints in all horses was 80.0%. However, the per-class classification rate was 21.5% for OA joints (sensitivity) and 99.1% for the other class (specificity). When the possible effect of class imbalance on the model performance was investigated by re-weighting the observations with the inverse of their frequency, the sensitivity of detection for OA joints was increased (61.1%) at the expense of specificity (81.9%), while the accuracy remained little changed (79.0%). The accuracies for all the classification tasks considered are shown in Table 1.

## 4. Discussion

This study assessed the utility of Fourier transform IR spectroscopy and spectral analyses using random forests to differentiate synovial fluid of horses with experimentally induced OA from controls. Good accuracy was obtained in discriminating synovial fluid of joints with induced OA from Sham-operated joints and OA joints from contralateral unoperated joints. This suggests that the response to the surgical creation of the osteochondral fragment was a key variable responsible for alterations in the spectral patterns of synovial fluid. Further supporting this is the lower accuracy obtained in differentiating Sham from Sham Control joints, indicating that the Sham operation does not cause marked changes in joint metabolism and homeostasis compared to contralateral unoperated controls for IR-based techniques. In agreement with these findings, Frisbie et al. (2008) [26] used the same experimental model and found that the increase in soluble biomarkers associated with the carpal chip model was greater than that associated with exercise alone in their control group. 

The classification rate by day of sampling was good. Overall, differentiation between horses with and without OA using IR spectroscopy and random forest classification was greater for SF than for serum in the same cohort of horses [36]. This finding is consistent with those reported for previous IR studies in dogs [33,45], although canine synovia fluid and serum testing were performed on samples from clinical cases. Synovial fluid is likely to reflect joint metabolism and disease more closely and specifically than serum due to the higher concentration of joint biomarkers in SF [26,46,47]. 

The accuracy of the classification model for identifying SF samples from OA joints in the current experimental study is similar to the value previously reported using IR techniques on synovial fluid to differentiate horses with osteochondrosis from controls (77%) [17] but lower than for horses affected by naturally occurring TA (89–97%) [16]. The difference in accuracy between these published studies could be attributable to the nature and chronicity of the two conditions or distinctive characteristics of the populations studied. Developmental diseases like osteochondrosis may have a lower burden and shorter disease duration than TA. In the TA study, the authors speculated that misclassification of some diseased joints as controls could have been due to the variation in duration and degree of inflammation, with joints with more moderate osteoarthritis perhaps not differing significantly from controls. The progressive and stepwise progression of clinically evident osteoarthritis and the intermittent expression of biomarkers often not proportionate to the burden of disease are well-recognized challenges to studying biomarkers in patients [6]. Controlled models such as the one used in the current study seek to overcome some of these challenges in the study of early disease. Nevertheless, the osteochondral fragment model used in our study may not have caused sufficient pathological changes in joint metabolism to affect the IR spectral profiles of SF to the same degree as those associated with naturally occurring TA. The low limit of detection for Fourier transform spectroscopy remains a challenge with the technology [48], which for some diseases has been overcome by processing samples to increase the concentration of known markers of interest [49]. Recent advances in ultra-broadband quantum IR spectroscopy hold promise for devices of increased sensitivity [50], as recently demonstrated for plasma from horses with osteoarthritis [51].

The clinical signs associated with the current carpal osteochondral model in these young untrained horses were relatively mild, in agreement with other studies using the same model [22,26]. In naturally occurring canine OA, IR spectroscopy on SF showed higher accuracy in differentiating diseased joints from controls compared with the current study [33]. A critical difference between the two studies, apart from the species of interest, was the superior performance of IR spectroscopy in these dogs in contrast with the induced disease in horses. Cranial cruciate ligament rupture in dogs is believed to be most commonly caused by prior ligament degeneration [52,53] and to have immune-mediated components [54,55], which could significantly contribute to functional groups associated with globular proteins within the IR profiles. Also, it is notable that the dogs recruited in that study by Malek et al. (2020) [33] had sufficiently advanced clinical OA and marked joint instability to warrant surgical correction. The short duration of our study may be insufficient to result in significant pathological changes affecting the SF IR spectral profiles compared with those measured in the SF from naturally diseased horses with more advanced OA.

The use of ELISA techniques on SF has proven successful for differentiating equine joints with induced OA using the same carpal osteochondral fragment model used in the current study [26]. More recently, another study reported excellent accuracy in discriminating SF of joints with induced OA from controls using ELISA techniques targeting specific markers (BAP, C2C, C12C, CPII, CS846, and CTXII) in an equine metacarpophalangeal osteochondral fragment model of OA [14]. The differences in accuracy compared with those studies are likely due to the sensitivity of the laboratory techniques chosen. These ELISA-based studies target the detection and quantification of specific known markers, while IR-based methods investigate changes in a broad array of IR-active molecules encompassing known and unknown biomarkers [29]. Interarticular variation in SF biochemical composition has been shown for normal equine joints [56,57]. Therefore, direct comparison with IR methods would require an IR-based evaluation of SF from the equine metacarpophalangeal osteochondral fragment model of OA.

## 5. Conclusions

This is the first study to evaluate the use of IR spectroscopy on synovial fluid from horses with experimental traumatically induced OA. This approach can differentiate affected joints from controls with good accuracy in a research setting. However, it does not compare as favorably to results from studies of naturally occurring joint disease in which clinical signs are present. Recent technical advances in the sensitivity of IR spectroscopy may contribute to improved disease detection using this technology in the future [48,50]. 

## Figures and Tables

**Figure 1 animals-14-00986-f001:**
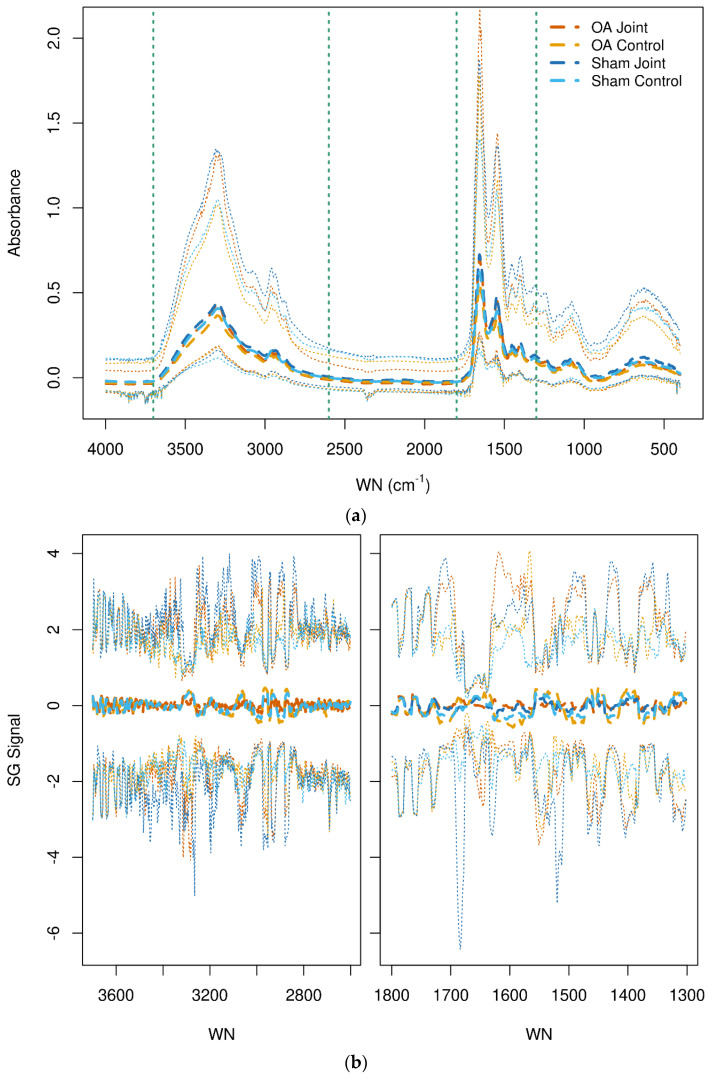
(**a**,**b**) Raw (top) spectra of synovial fluid from all joints. The vertical dashed lines indicate the wavenumber fingerprint regions 3700–2600 cm^−1^ and 1800–1300 cm^−1^. Savitzky-Golay transformed (bottom) spectra of synovial fluid from all joints in the selected fingerprint regions. The median (thick dashed lines) and 2.5% and 97.5% quantiles (thin dotted lines) are shown. The osteoarthritis (OA) and OA Control joint spectra are shown in dark and light orange, respectively. The Sham and Sham Control joints are shown in dark and light blue, respectively. WN = wavenumber.

**Table 1 animals-14-00986-t001:** Accuracy of the prediction models by classification task.

Comparison Task		No. of Classes	Prediction Accuracy (%)
1	Day	10	87
2	OA vs. OA Control joints	2	75
3	Sham vs. Sham Control joints	2	61
4	OA joint vs. Sham joint	2	70
5a	Joint group (OA vs. OA Control vs. Sham vs. Sham Control)	4	53
5b	OA joint vs. any other joint	2	80
6	Horse group (OA vs. Sham)	2	68
7a	Day × joint group	37	38
7b	Day × OA joint	9	67
8	Horse sampled among all horses	17	46

OA = osteoarthritis.

## Data Availability

Data is available from the corresponding author on reasonable request.

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
