# Peer review of "Infrared Spectroscopy of Synovial Fluid Shows Accuracy as an Early Biomarker in an Equine Model of Traumatic Osteoarthritis"

_animals, 2024, doi:10.3390/ani14070986_

Round 1

Reviewer 1 Report

Comments and Suggestions for Authors

Dear, Authors

My comments are below:

In the introduction, it is said that the technique of IR spectroscopy has performed well in SF from horses with naturally occurring diseases, but the reference was not included (line 95).

It is not clear why to carry out the same study with experimentally induced OA if the research has already done successfully with naturally occurring joint disease (line 97). The authors could present more arguments that justify why to induce injuries (Describe what information would be added with experimental OA model research that was not obtained in the previous publication).

Materials and methods: The anesthetic protocol described did not include sedation (line 123).

The results on lameness, flexion tests, effusion and radiographic scores that confirm the establishment of OA were not presented, suggesting that the reader turn to another publication to access this information (line 158). It would be more convenient if the reader could get all the information about the research in the article itself.

In the discussion, the authors compared results that were not presented in the article (line 290). If the results obtained from serum and SF analyzes were presented in a single article, the publication would be much more interesting.

It was concluded that “A more extended study period or a more invasive equine model of joint injury may be required to explore the full potential of IR spectroscopy as a diagnostic technique”. However, the only justification presented for the use of the experimental model was the possibility of “early detection of the OA”, which would not justify prolonging the study period. Furthermore, proposing “a more invasive equine model of joint injury” means saying that the model adopted was not efficient in inducing OA, which is controversial with the statements made in the study itself, which confirmed the establishment of OA.

Author Response

Dear Reviewer

Thank you for your considered feedback. Our responses are listed below, and related revisions are tracked in the revised version of the manuscript. You will note other changes have been made that were directed by the editor and another reviewer. Cheers.

Abstract

  • Provides rationale for study and appropriate back ground
  • Summary of samples and groups- what were the ages + or – SEM between groups.

The median ages of the horses have been added (data are not normally distributed).

  1. Introduction
  • Comprehensive introduction, outlining background, the problem and potential solution. Subsequent reasoning for study.
  1. Materials and Methods
  • Provide more detail RE donors – age between groups

Age data were not normally distributed; the median and interquartile range of the horses’ ages have been added.

  • What was the rationale behind the total number of horses included in the study?

The rationale is given in the materials and methods section, first paragraph.

  • Was SF processed – spun to remove cellular debris and frozen in liquid nitrogen prior to storing at -80?

No. Although not published, the authors previously conducted a small laboratory trial in which ultracentrifugation was performed on split samples, and spectra of the resuspended cellular pellet supernatant, and neat fluid were collected. The latter provided the most spectral information. 

  • Was SF Hyalaronidase treated before analysis?

No. Although this has value in the separation of proteoglycans and glycosaminoglycans in traditional liquid chromatography, IR spectroscopy does not rely upon molecular separation. Indeed, frequency dependent spectra arise from IR absorption by the bending of and stretching of IR active molecular bonds. The chemical alteration associated with the use of hyaluronidase would likely change the spectra. It would also introduce a step that is less financially and time-saving than placing the neat fluid on the optical window of the spectrometer.   

  1. Discussion/ Conclusion
  • What is defined as ‘good accuracy’?

The authors have defined this in the materials and methods section of the paper.

  • First to do in SF but other groups have looked at this technique and similar for biomarkers in equine OA- https://doi.org/10.1039/D2AY00779G , https://doi.org/10.1111/evj.13115

The authors are familiar with the work of Paraskevaidi et al. (2020) on serum in preparing this manuscript on FTIR of synovial fluid but were not aware of the work by Clarke et al. (2022) in plasma– thank you. We have now referred to the work of Clarke et al. (2022), as an example of a QCL based spectrometer. Please note, neither study explored synovial fluid.

Reviewer 2 Report

Comments and Suggestions for Authors

1.       Abstract

·         Provides rationale for study and appropriate back ground

·         Summary of samples and groups- what were the ages + or – SEM between groups.

2.       Introduction

·         Comprehensive introduction, outlining background, the problem and potential solution. Subsequent reasoning for study.

3.       Materials and Methods

·         Provide more detail RE donors – age between groups

·         What was the rationale behind the total number of horses included in the study?

·         Was SF processed – spun to remove cellular debris and frozen in liquid nitrogen prior to storing at -80?

·         Was SF Hyalaronidase treated before analysis?

4.       Discussion/ Conclusion

·         What is defined as ‘good accuracy’?

·         First to do in SF but other groups have looked at this technique and similar for biomarkers in equine OA- https://doi.org/10.1039/D2AY00779G , https://doi.org/10.1111/evj.13115

Author Response

Dear Reviewer

Thank you for your considered feedback. Our responses are listed below, and related revisions are tracked in the revised version of the manuscript. You will note other changes have been made that were directed by the editor and another reviewer. Cheers.

Dear, Authors

My comments are below:

In the introduction, it is said that the technique of IR spectroscopy has performed well in SF from horses with naturally occurring diseases, but the reference was not included (line 95).

Respectfully, yes it is. However, more details on the study on traumatic arthritis cases is provided.

It is not clear why to carry out the same study with experimentally induced OA if the research has already done successfully with naturally occurring joint disease (line 97). The authors could present more arguments that justify why to induce injuries (Describe what information would be added with experimental OA model research that was not obtained in the previous publication).

The authors have added more details on the justification for the study and references that support this approach.

Materials and methods: The anesthetic protocol described did not include sedation (line 123).

Thank you. Now added.

The results on lameness, flexion tests, effusion and radiographic scores that confirm the establishment of OA were not presented, suggesting that the reader turn to another publication to access this information (line 158). It would be more convenient if the reader could get all the information about the research in the article itself.

The authors agree. However, copyright rules and the requirements of this journal preclude republishing these data. 

In the discussion, the authors compared results that were not presented in the article (line 290). If the results obtained from serum and SF analyzes were presented in a single article, the publication would be much more interesting.

Respectfully, the results discussed in comparison here are those published by Frisbie et al. (2008); not our own. Regarding the reviewer's second comment, the methodology used in the spectral analyses is different from those used for SF samples.

It was concluded that “A more extended study period or a more invasive equine model of joint injury may be required to explore the full potential of IR spectroscopy as a diagnostic technique”. However, the only justification presented for the use of the experimental model was the possibility of “early detection of the OA”, which would not justify prolonging the study period. Furthermore, proposing “a more invasive equine model of joint injury” means saying that the model adopted was not efficient in inducing OA, which is controversial with the statements made in the study itself, which confirmed the establishment of OA.

Agreed, and omitted.